# Long COVID: A Systematic Review of Preventive Strategies

**DOI:** 10.3390/idr17030056

**Published:** 2025-05-21

**Authors:** Sun O. Park, Neha Nanda

**Affiliations:** Division of Infectious Disease, Department of Medicine, Keck School of Medicine, University of Southern California, Los Angeles, CA 90033, USA

**Keywords:** long COVID, post-acute sequelae of COVID-19 (PASC), post-acute COVID-19 syndrome (PACS), post-COVID-19 conditions (PCCs), COVID-19, prevention

## Abstract

**Background:** Since the emergence of severe acute respiratory syndrome coronavirus 2 (SARS-CoV-2) in December 2019, long COVID (LC) has become a significant global health burden. While knowledge about LC is accumulating, studies on its prevention are still lacking. **Methods:** We conducted a systematic review following the PRISMA (Preferred Reporting Items for Systematic Reviews and Meta-Analyses) guidelines to investigate prevention options for LC. We identified fifteen articles on vaccines, seven on antivirals, and six on other interventions after searching for articles in the PubMed/MEDLINE database using the MeSH terms. **Results:** Most vaccine-related studies demonstrated a protective effect of COVID-19 vaccines against developing LC. Our review found an equivocal effect of antivirals, while metformin had a protective effect in outpatients and corticosteroids were protective in hospitalized patients against LC. Conversely, COVID-19 convalescent plasma and multiple micronutrient supplement did not confer any protection against LC. **Conclusions:** COVID-19 vaccination is vital as it not only prevents COVID-19 but also reduces the severity of illness and may help prevent LC. Further studies are warranted to shed light on preventive strategies for long COVID.

## 1. Introduction

Severe acute respiratory syndrome coronavirus 2 (SARS-CoV-2), the causative agent of COVID-19, was first identified in December 2019 in the city of Wuhan, China. COVID-19 rapidly spread worldwide, resulting in a major global pandemic [1] that has caused over 1 million deaths in the United States (US) and 7 million deaths globally [2]. Moreover, a subset of individuals infected with SARS-CoV-2 may experience persistent somatic symptoms, a condition known as long COVID (LC), also referred to as Post-Acute Sequelae of COVID-19 (PASC), Post-Acute COVID-19 Syndrome (PACS), or Post-COVID-19 Conditions (PCCs) [2,3].

LC is a complex and heterogeneous condition lacking specific biomarkers, which has posed challenges in accurately estimating its burden and developing prevention and management strategies [2,3]. While definitions of LC vary, international scientific agencies agree on a timeline of symptom persistence, defining it as symptoms lasting 3 months or longer. The World Health Organization (WHO) defines LC as the continuation or development of new symptoms—including over 200 different symptoms that can impact everyday functioning—occurring 3 months after the initial SARS-CoV-2 infection, with no other explanation [4]. Similarly, the US National Academic Sciences Engineering Medicine (NASEM) described LC as a SARS-CoV-2 infection-associated condition persisting beyond 3 months, presenting as a continuous, relapsing and remitting, or progressive disease state affecting one or more organ systems [3]. The Centers for Disease Control and Prevention (CDC) updated its definition to reflect symptoms persisting for at least 3 months after infection.

Based on data collected between 2020 and 2021 from 22 countries, approximately 6.2% of individuals of all ages experienced at least one of three LC symptom clusters—persistent fatigue with body pain or mood swings, cognitive problems, or ongoing respiratory issues—3 months after SARS-CoV-2 infection [5]. LC negatively impacts quality of life, leading to significant health and economic strain on individuals and health care systems [6].

Studies demonstrating the impact of vaccines or antivirals on LC are accumulating; however, systematic reviews synthesizing these outcomes are scarce. A recent meta-analysis, in which LC was defined as symptoms lasting 30 days or longer, found that antiviral treatment during acute COVID-19 significantly reduced the risk of LC; in contrast, corticosteroids and monoclonal antibodies did not show protective effects [7]. Another meta-analysis assessing vaccines’ effect found that receiving a primary vaccine series prior to COVID-19 was associated with reduced odds of developing LC, while one dose or three does did not [8]. A meta-analysis of heterogeneous studies may lead to erroneous conclusions. We aim to shed light on interventions for the prevention of LC. Our objectives are to (1) review the pathogenic mechanisms and risk factors of LC and (2) systematically review interventions, particularly vaccines and antivirals, for LC prevention using the definition of LC as symptoms persisting for 3 months or longer after SARS-CoV-2 infection, aligning with the definitions of the WHO, NASEM, and CDC.

## 2. The Pathogenic Mechanism of Long COVID

Long COVID can affect multiple organs, including those in the respiratory (e.g., cough or shortness of breath), cardiovascular (e.g., chest pain or palpitations), neurological (e.g., headache, loss of taste or smell, anxiety, cognitive impairment, fatigue, or insomnia), gastrointestinal (e.g., nausea, vomiting, diarrhea, abdominal pain, or loss of appetite), endocrine (e.g., thyroid dysfunction or new onset diabetes), musculoskeletal (e.g., myalgia or joint pain), reproductive (e.g., erectile dysfunction or irregular menstruation), and immune systems (e.g., autoimmunity or mast cell activation syndrome), leading to a wide variety of symptoms [3,9,10,11]. Long-lasting symptoms after acute infection with SARS-CoV-2 are not unique; a variety of chronic conditions triggered by other infectious agents have been well described [3]. Examples include long-lasting symptoms caused by neurotrophic herpesviruses, enteroviruses, Ebola virus, MERS viruses, *Borrelia* spp. (implicating pathogen for Lyme disease), and chronic fatigue syndrome caused by *Coxiella burnetii* (implicating pathogen for Q fever) [3]. Moreover, there are many similarities in the pathophysiology and biological abnormalities between myalgic encephalomyelitis/chronic fatigue syndrome (ME/CFS) and LC, including nervous system dysfunction, cardiovascular abnormalities, immune system dysregulation, metabolic alterations, and gut microbiome alterations, suggesting potential overlapping mechanisms [12]. Insights from one condition may help understand the other [12]. At this time, the mechanisms of LC pathogenesis are not fully elucidated and can vary between individuals [3] and due to other external factors. For example, the incidence of LC may be less common with Omicron variants compared to earlier variants [13]. Interestingly, compared to the pre-Delta and Delta eras combined, gastrointestinal, metabolic, and musculoskeletal disorders have increased during the Omicron era, while symptoms related to other systems like the respiratory and neurological systems have decreased over time. This indicates a shift in the phenotypic features of LC, which is likely influenced by the evolving virus and vaccination [13].

The key mechanisms hypothesized for LC pathogenesis include persistent viral reservoirs, immune dysregulation and inflammatory responses, autoimmunity, reactivation of latent herpesviruses, microvascular and endothelial dysfunction, nervous system dysfunction, and microbiota dysbiosis.

Persistent viral reservoirs: LC consists of a broad range of symptoms lasting beyond 3 months after infection regardless of the presence of viral particles. However, compared to individuals who do not develop LC, LC patients are more likely to have persistent viral RNA or protein in their tissues or blood, potentially establishing a reservoir [14,15,16]. In one of the studies, autopsies on 44 patients were conducted, and in 11 patients, they detected the replication of SARS-CoV-2 in multiple anatomic sites, including the brain [15]. Additionally, SARS-CoV-2 proteins or RNA in specific tissues may drive LC pathogenesis, such as chronic inflammation, immune dysregulation, latent pathogens reactivation, vascular abnormalities, and microbiome dysbiosis, leading to persistent symptoms [14].

Immune dysregulation and inflammatory responses: Insufficient SARS-CoV-2 antibody production and altered T cell responses during acute COVID-19 may be predictive of LC [9,17,18]. LC patients demonstrate persistently elevated inflammatory cytokines, such as IFN- β, IFN- λ1, IL-1β, IL-6, TNF-α, IP10, and CCL11, with some remaining elevated even months after COVID-19 infection [18,19]. In addition, LC patients exhibit decreased CD4+ and CD8+ effector memory cells, highly activated innate immune cells, and a loss of naïve T and B cells, possibly reflecting the persistent conversion of naïve T cells into activated phenotypes driven by bystander activation or prolonged antigen presentation due to underlying inflammation [19]. The increased expression of T cell exhaustion markers (PD-1 and TIM-3) and activation markers (CD38 and HLA-DR), together with the above immune dysregulation, may contribute to persistent symptoms in LC patients by impairing viral clearance and promoting chronic inflammation [19]. The production of autoantibodies (autoAbs), such as anti-nuclear Abs (Ro/SS-A, La/SS-B, U1-snRNP, Jo-1, and P1) and anti-IFN-a2, is common in LC patients, which is negatively correlated with the production of anti-SARS-CoV-2 antibodies [20]. Furthermore, some individuals may already have autoAbs even before COVID-19 infection, and COVID-19 may unmask subclinical autoimmune conditions, leading to persistent inflammation and immune changes that contribute to LC [20]. Additionally, immune dysregulation promoted by SARS-CoV-2 can reactivate latent herpesviruses such as EBV and CMV, potentially worsening LC symptoms [9,11,12,21].

Microvascular and endothelial dysfunction: SARS-CoV-2 can impair microcirculation and endothelial function, leading to reduced oxygen delivery, clotting, and subsequent multiorgan damages [9,10]. Persistent microclots containing inflammatory molecules, such as α2-antiplasmin and amyloid A, have been observed in LC patients, potentially contributing to persistent symptoms [22].

Nervous system dysfunction: Neurological symptoms in LC patients may result from the direct neuroinvasion of SARS-CoV-2 through crossing the blood–brain barrier, accessing neural pathways, or from neuroinflammation triggered by systemic inflammatory responses caused by immune cell activation and cytokine release [9,23,24,25]. Additionally, impaired peripheral vasoconstriction and reduced tissue oxygen supply may contribute to neurological dysfunction [24,25]. Key characteristics of LC include neurological and cognitive dysfunction (“brain fog”), such as fatigue, headache, memory loss, cognitive impairment, sensorimotor dysfunction, altered smell and taste, and autonomic dysfunction such as temperature dysregulation, postural orthostatic tachycardia syndrome (POTS), and orthostatic hypotension [9,12,25].

Microbiota dysbiosis: SARS-CoV-2 can disrupt the gut microbiome, resulting in persistent gastrointestinal (GI) symptoms and even respiratory and neurological symptoms [25,26,27,28]. The restoration of healthy microbiota could potentially reverse chronic symptoms in LC patients [29].

## 3. Risk Factors for Long COVID

Multiple studies have shown that females are more likely than males to develop LC [30,31,32,33]. A prospective cohort study found that the female sex was associated with a 3-fold higher risk of developing LC [31]. A multicenter cohort study also described that the number of LC symptoms was significantly higher in females than in males, with means of 2.25 and 1.5, respectively [32]. Biological factors, such as higher estrogen levels in females, might contribute to the development of LC, as estrogen can upregulate angiotensin-converting enzyme 2 (ACE2), a receptor used by SARS-CoV-2 to enter host cells [10,34], although the impact of ACE2 on LC remains unclear [34]. Behavioral differences, such as females being more attentive to bodily distress and more likely to seek medical care, may also contribute to the higher rate of LC in females [31,32].

Another risk factor for LC is the severity of the illness during the acute phase. Patients with more severe symptoms—such as being hospitalized, requiring analgesics, or needing additional treatments, are at a higher risk of developing LC [35,36,37]. For example, a population-based cohort study with a total of 1459 patients found that the prevalence of LC was higher in hospitalized patients than in non-hospitalized patients, with prevalence rates of 72.6% and 46.2%, respectively [35]. Additionally, a cohort study by Guzman-Esquivel et al. demonstrated a positive relationship between LC and factors such as hospitalization and antibiotic use [36].

Middle age or advanced age is associated with an increased risk of developing LC [37,38]. Additionally, overweight and obesity have been linked to a higher risk of developing LC [37,39]. A high overall burden of comorbidities, as well as pulmonary conditions such as asthma, chronic obstructive pulmonary disease, and obstructive sleep apnea, are also positively associated with LC [35,39,40]. Furthermore, pre-infection psychological distress significantly increases the risk of LC [41]. In a large prospective cohort study using data from Nurses’ Health Study (predominantly female cohorts), individuals experiencing two or more types of distress prior to infection had nearly a 50% higher risk of developing LC [41].

## 4. Materials and Methods

A systematic review was conducted following the PRISMA (Preferred Reporting Items for Systematic Reviews and Meta-Analyses) guidelines [42] to evaluate prevention strategies for LC, including vaccines and antivirals. This study is registered on the International Platform of Registered Review and Meta-analysis Protocols (INPLASY) with the registration number INPLASY202530002. Two reviewers (SP and NN) independently assessed the quality and risk of bias of the studies, and any discrepancies were resolved through discussion.

LC was defined as persistent symptoms lasting three or more months and affecting one or more organ systems after SARS-CoV-2 infection, encompassing the definitions outlined by the WHO, NASEM, and CDC [3]. We searched the PubMed/MEDLINE database for articles published between December 2019 and September 2024 using the following MeSH terms: “Post-Acute COVID-19 Syndrome (PACS)” AND “prevention and control”, “PACS” AND “Vaccines”, “PACS” AND “remdesivir (Gilead Sciences, Inc., Foster City, CA, USA)”, “PACS” AND “nirmatrelvir (Pfizer Inc., New York, NY, USA )”, and “PACS” AND “molnupiravir (Merck & Co., Inc., Rahway, NJ, USA)”.

We included articles that met the following criteria: (1) study population consisting of adults aged 18 years or older, (2) LC being defined as described above, and (3) individuals who were categorized as vaccinated or treated received vaccines or treatment before the onset of LC. We excluded the following studies: (1) studies that either did not define LC or defined LC with symptoms lasting less than 3 months, (2) studies that did not clearly indicate the timeline of vaccinations or treatment, and (3) studies that did not report a statistical significance value. For the purpose of this review, we did not include case reports or series, review articles, and meta-analyses.

Based on the above criteria, we included 28 studies [43,44,45,46,47,48,49,50,51,52,53,54,55,56,57,58,59,60,61,62,63,64,65]. Of these, fifteen focused on the impact of COVID-19 vaccination on LC, seven studied the impact of antivirals on LC, and six investigated the impact of other treatments on the development of LC (Figure 1).

Additionally, we compared the impact of COVID-19 vaccines and antivirals with those when LC was defined as persistent symptoms lasting 28 days or longer, as there are a significant number of studies with this definition.

## 5. Results

Unless specified otherwise, the findings refer to LC being defined as persistent symptoms lasting 3 months or longer.

### 5.1. Impact of COVID-19 Vaccine on Prevention of Long COVID

We identified fifteen articles focused on the impact of COVID-19 vaccination on LC. Unless described by the authors, COVID-19 infection before January 2021 is categorized as wild type, infection between January 2021 and June 2021 as the Alpha variant, infection between July and December 2021 as the Delta variant, and infection from January 2022 onwards as the Omicron variant (Table 1). We specified vaccine types when the information was available (Table 1).

Of the fifteen articles, ten studies demonstrated a protective effect of vaccines against the development of LC (Table 1). Among five studies [45,48,57,66,67] that did not demonstrate a statistically significant protective effect on the development of LC, one study showed a trend towards protection [67], and two reported a reduction in a few LC symptoms due to vaccination [45,48]. In this section, we describe the identified prospective studies in detail and summarize identified retrospective and cross-sectional studies, focusing on the impact of vaccines on the development of LC. A summary of each selected study is described in Table 1.

Ayoubkhani et al. [44] conducted a prospective cohort study in the United Kingdom (UK) to determine the trajectory of LC in participants who were vaccinated after being diagnosed with COVID-19. The study included 28,356 participants who had at least one visit between February and September 2021 (Alpha and Delta variants), a test confirming COVID-19 diagnosis at least 12 weeks before the final visit, and had received a vaccine after the diagnosis of COVID-19 but before the above study period. The odds of developing LC decreased by 12.8% after the first vaccine dose (BNT162b2, mRNA-1273, or ChAdOx1) compared to unvaccinated individuals and then decreased by an additional 8.8% after the second vaccine dose.

Antonelli et al. [66] investigated the effect of a third dose of (booster) of a monovalent vaccine compared to a second dose during the Delta and Omicron eras, separately, through a prospective community-based case–control study in the UK. The study included COVID-19 patients from June 2021 to April 2022, with 1910 patients in each group during the Delta period and 7894 in each group during the Omicron period. The booster vaccination did not reduce the risk of LC when it was defined as symptoms persisting for 12 weeks or longer despite a trend towards a protective effect. The booster vaccination reduced the risk of LC when LC was defined as symptoms persisting for 4 weeks or longer during the Delta period, but not during the Omicron period.

A prospective longitudinal cohort study was carried out by Richard et al. [67] in the USA using data from the Epidemiology, Immunology, and Clinical Characteristics of Emerging Infectious Diseases with Pandemic Potential (EPICC) study. The study population consisted of 1832 military personnel aged 18 to 44 years who tested positive for SARS-CoV-2 from February 2020 to December 2021 (wild type and Alpha and Delta variants). Fully vaccinated individuals (two doses of BNT162b2 or mRNA-1273 or one dose of Ad26.COV2.S) were 27% less likely to develop LC, though the difference was not statistically significant (*p* = 0.17). However, when LC was defined as symptoms persisting 4 weeks, the risk was reduced by 28% with statistical significance (*p* = 0.02).

Nehme et al. [59] compared the prevalence of LC symptoms between individuals infected with COVID-19 (Omicron variants BA.1 and BA.2) (total 1807) and uninfected individuals (total 882), as well as between fully vaccinated (at least two doses of mRNA-1273 or BNT162b2 before COVID-19) and unvaccinated individuals. This study included individuals who were diagnosed with Omicron infection between December 2021 and February 2022 at the Geneva University Hospitals outpatient testing center and were contacted 3 months from their test date. Overall, 11.7% of individuals infected with Omicron had LC symptoms compared to 10.4% of individuals who tested negative (*p* < 0.001). LC symptoms were less common in fully vaccinated individuals compared to unvaccinated individuals (adjusted prevalence of 9.7% vs. 18.1%; *p* < 0.001)

A prospective cohort study by Brunvoll et al. [48] included 1420 participants diagnosed with COVID-19 between November 2020 and October 2021 (wild type and Alpha and Delta variants) in Norway. The study found no differences in self-reported LC symptoms between fully vaccinated (at least two doses of mRNA) and unvaccinated individuals, except vaccinated individuals reported fewer memory problems. However, the study found no differences in Everyday Memory Questionnaire-13 (EMQ-13) scores between vaccinated and unvaccinated individuals.

Abu Hamdh et al. [43] carried out a prospective cohort study in Palestine, including 669 patients diagnosed with COVID-19 between September 2021 and January 2022 (Delta variant), of whom 6.7% required hospitalization. A total of 41% patients received at least one dose of vaccine (BNT162b2, Sputnik Light, mRNA-1273, Sputnik V, or ChAdOx1) before their COVID-19 diagnosis. Vaccinated patients with at least one dose had 85% lower odds of developing LC compared to unvaccinated patients.

Another prospective study in Pakistan, including a total of 481 hospitalized patients with COVID-19 between February and June 2021 (Alpha variant) [54], showed that fully or partially vaccinated patients were less likely to develop LC compared to unvaccinated patients, with adjusted odds ratios (aORs) of 0.38 (95% CI: 0.20–0.70) and 0.44 (95% CI: 0.24–0.80), respectively. Similarly, a prospective study conducted in Brazil involving 412 hospitalized patients diagnosed with COVID-19 between May 2021 and February 2022 (Alpha, Delta, and Omega variants) [68] demonstrated that being fully vaccinated (at least one dose of Janssen or two doses of other vaccines) reduced the odds of developing LC (aOR = 0.55, *p* = 0.007).

A staggered retrospective cohort at the population level was carried out by Català et al. [49] using primary care records from the United Kingdom (UK) (Clinical Practice Research Datalink [CPRD] GOLD and CPRD AURUM), Spain (the Information System for Research in Primary care [SIDIAP]), and Estonia (CORIVA database), which included over 10 million vaccinated and over 10 million unvaccinated people. Analyses included both patients infected with COVID-19 and uninfected patients to assess vaccines’ effect in preventing COVID-19 and subsequent LC. Individuals were registered by January or February 2021 and followed up through December 2022 (Alpha, Delta, Omicron variants), and vaccination status was staggered based on the vaccine rollout period. COVID-19 vaccination (at least one dose of BNT162b2, ChAdOx1, mRNA-1273, or Ad26.COV2.S) showed a significant protective effect against developing LC, with overall vaccine effectiveness ranging from 29% to 52%. Trinh et al. [61] conducted the same analyses as Català et al. [49] using the Norwegian Linked Health Registries at University of Oslo, which covers the entire Norwegian population of approximately 5.4 million, and demonstrated a reproducible and significant protective effect of vaccines against developing LC with 36% effectiveness.

Luo et al. [57] performed a retrospective cohort in Hong Kong using electronic and telephone follow-up records for COVID-19 patients diagnosed between December 2021 and May 2022 (predominantly Omicron variant), with a follow-up of 6 to 12 months. They found that receiving a booster vaccine (three or more doses of BNT162b2 or CoronaVac) did not reduce the risk of LC compared to fewer doses before COVID-19. However, vaccination after infection increased the risk of reporting LC symptoms, which may be due to side effects from vaccination. MacCallum-Bridges et al. [58] conducted a population-based retrospective cohort study, using data from the Michigan COVID-19 Recovery Surveillance Study for participants diagnosed with COVID-19 between March 2020 and May 2022 (wild type and Alpha, Delta, and Omicron variants), which found a 58% lower adjusted prevalence of 90-day LC in the fully vaccinated group (two doses of BNT162b2, mRNA-1273, ChAdOx1, or Sinovac or one dose of Ad26.COV2.S) compared to the unvaccinated group. Babicki et al. [45] conducted a retrospective study in Poland using data from the STOP-COVID registry (unknown study period), with 9.4% individuals being vaccinated before COVID-19 and 73.6% after. Full vaccination (two doses of BNT162b2, mRNA-1273 or ChAdOx1 or one dose of Ad26.COV2.S) had no impact on the development of LC, although headache, arthralgia, and a new diagnosis of hypertension were less common in vaccinated individuals regardless of the timeline of vaccination.

Woldeglorgis et al. [63] assessed LC prevalence 90 days post COVID-19 in highly vaccinated Western Australian adults between July and August 2022 (Omicron variant). Those with 0–2 vaccine doses had a 1.4-fold higher risk, and those with 3 doses had a 1.3-fold higher risk of developing LC compared to individuals with 4 or more doses. Xie et al. [64], using data from the 2022 National Health Interview Survey of the United States adults (Omicron variant), showed that individuals who received a booster dose had 25% lower odds of developing LC compared to unvaccinated individuals.

Among eleven studies [13,44,69,70,71,72,73,74,75,76,77] that defined LC as persistent symptoms lasting 28 days or longer, ten studies, including two prospective studies [70,72] and eight retrospective studies [13,44,69,71,73,74,75,76], showed a protective effect of COVID-19 vaccines (Appendix A).

### 5.2. The Impact of Antivirals on the Prevention of Long COVID

A total of seven studies met our inclusion criteria under this section. We identified one randomized trial, three prospective studies, and three retrospective studies. A summary of each study is described in Table 2.

Among the three studies that evaluated the impact of remdesivir on the prevention of LC in hospitalized patients, a randomized trial [60] found no differences between remdesivir-treated and untreated groups, while a prospective study [78] and a retrospective case–control study [55] demonstrated that remdesivir had a protective effect, as described below.

Nevalainen et al. [60] reported a one-year follow-up of the randomized SOLIDARITY Finland trial on the effects of remdesivir on recovery and LC symptoms. Among 181 patients recruited from 11 Finnish hospitals between July 2020 and January 2021 (wild type) who completed the 1-year survey, no significant differences were found in self-reported recovery or LC symptoms between remdesivir-treated and untreated groups. A prospective study conducted by Boglione et al. [78] included 163 remdesivir-treated and 165 untreated patients who were hospitalized for COVID-19 during the March 2020-January 2021 period (wild type and the Alpha variant) at Saint Andrea hospital in Vercelli, Italy. The study showed a 36% reduction in the odds of developing LC in the remdesivir-treated group. A case–control study by Fernández-de-las-Peñas et al. [55] included hospitalized COVID-19 patients from urban hospitals in Madrid, Spain, between September 2020 and March 2021 (wild type and the Alpha variant). The odds of developing LC symptoms following acute COVID-19 infection was 60% lower in patients treated with remdesivir during their acute infection compared to the patients who were not treated with remdesivir.

Among the three studies that evaluated the impact of nirmatrelvir/ritonavir on the prevention of LC, a prospective study [62] demonstrated a protective effect, while the other prospective cohort [53] and a retrospective cohort [51] did not.

A prospective cohort study conducted by Durstenfeld et al. in the United States (US) [53] involved patients diagnosed with COVID-19 between March and August 2022 (Omicron variant), utilizing data from the COVID Citizen Science (CCS) study. This study, which included 353 patients treated with nirmatrelvir/ritonavir and 1258 untreated patients, demonstrated that nirmatrelvir/ritonavir did not reduce the risk of LC at 3 months or later post COVID-19 infection based on a propensity-adjusted model. Unlike the above study, Wang et al. [62] demonstrated a protective effect of nirmatrelvir/ritonavir in the development of LC in their prospective cohort study in China, which involved COVID-19 patients discharged between April and June 2022 (Omicron variant); nirmatrelvir/ritonavir treatment during hospitalization (178 out of 634 patients received nirmatrelvir/ritonavir) reduced the odds of developing LC by 65% 6 months after hospital discharge. In a retrospective cohort study by Congdon et al. including 500 COVID-19 patients in New York, USA, from May to November 2022 (Omicron variant) [51], nirmatrelvir/ritonavir was not associated with a significant reduction in the overall risk of developing LC.

Lastly, a retrospective study by Bertuccio et al. [46] showed a protective effect of antivirals, such as remdesivir, molnupiravir, or nirmatrelvir/ritonavir. Bertuccio et al. [46] included 649 outpatients with mild to moderate COVID-19 in Lombardy, Northern Italy, and had at least one risk factor for disease progression between April 2021 and March 2022 (Alpha, Delta, and Omicron variants). Patients treated with antivirals, including remdesivir, molnupiravir, and nirmatrelvir/ritonavir, had a 57% reduction in the odds of developing any symptoms compared to those not treated with antivirals. Additionally, patients treated with monoclonal antibodies (mAbs) had a 52% reduction in the odds of developing neuro-behavioral symptoms compared to those not treated with mAbs.

Among four retrospective studies [79,80,81,82] that defined LC as persistent symptoms lasting 28 days or longer, three studies [79,81,82] showed that nirmatrelvir or molnupiravir had a protective effect (Appendix A).

### 5.3. The Impact of Other Treatments on the Prevention of Long COVID

Six studies met the inclusion criteria under this section, including three randomized trials, one secondary analysis of a randomized clinical trial, and two retrospective cohorts. A randomized clinical trial in outpatients [56] and a secondary analysis of a randomized clinical trial in hospitalized patients [65] found no protective effect of COVID-19 convalescent plasma (CCP) on the development of LC. In contrast, a randomized trial by Bramante et al. [47] demonstrated that metformin reduced LC risk in outpatients, and two retrospective cohort studies [50,52] demonstrated a protective effect of corticosteroids in hospitalized patients. Lastly, multiple micronutrient supplements did not demonstrate a protective effect in a randomized trial [83]. A summary of each study is presented in Table 3.

Gebo et al. [56] conducted a double-blind multicenter randomized controlled trial involving 1225 symptomatic COVID-19 outpatients from June 2020 to October 2021 (wild type and Alpha and Delta variants) across 23 sites in the USA, with a 90-day follow-up after CCP transfusion. While there was no statistically significant reduction in the odds of self-reported LC symptoms at 90 days for those who received CCP within 5 days of illness compared to the control group, a trend towards lower odds of LC was observed. Yoon et al. [65] conducted a secondary analysis of a blinded, multicenter randomized controlled trial involving 281 hospitalized COVID-19 patients from April 2020 to March 2021 (wild type and Alpha variant), with an 18-month follow-up. The study found no significant differences in the odds of developing LC between the CCP and placebo groups.

Bramante et al. [47] carried out a multi-site, phase 3, randomized quadruple-blinded placebo-controlled trial to assess whether early treatment with metformin, ivermectin, or fluvoxamine could prevent LC in COVID-19 outpatients with overweight or obesity. Patients were enrolled within 3 days of COVID-19 diagnosis and within 7 days of symptom onset between December 2020 and January 2022 (wild type and Alpha, Delta, and Omicron variants). The metformin group showed a 42% reduction in LC incidence compared to the control, while ivermectin or fluvoxamine showed no effect.

Two retrospective cohort studies demonstrated that corticosteroid treatment lowered the risk of developing LC—one was conducted in the Netherlands and involved hospitalized COVID-19 patients between March 2020 and September 2021 (wild type and Alpha and Delta variants) with the criteria for corticosteroid use being a respiratory rate of >25 breaths per minute, an oxygen requirement of >5 L, or a rapidly increasing oxygen requirement [52], and another was conducted in Spain in hospitalized COVID-19 patients between March and May 2020 (wild type) [50]. The criteria for corticosteroid use were not mentioned in the study performed in Spain [50].

Two centers in Catalonia, Spain, investigated the effect of a multiple micronutrient supplement (MMS), comprising Vitamins A, B6, B12, C, D3, E, copper, folic acid, iron, selenium, and zinc, on LC incidence in a double-blind, placebo-controlled, randomized clinical trial. The study included 246 patients with mild COVID-19 (not requiring hospital admission) between September 2021 and February 2023 (Delta and Omicron variants). The trial found no differences in LC incidence between the intervention group (MMS for 2 weeks starting on day 1 of inclusion) and the placebo groups [83].

## 6. Discussion

Long COVID (LC) poses significant medical, social, and economic impacts worldwide [3]. The underlying mechanisms of LC remain elusive. The proposed mechanisms are not mutually exclusive and may interact with each other, leading to a complex and varied clinical presentation seen in patients with LC. The risk factors are not easily modifiable. Based on that, our systematic review focused on modifiable preventive strategies for LC.

Despite the numerous studies on LC, research on its prevention remains challenging due to the evolving nature of the virus, the lack of unique biomarkers for diagnosing LC, and variations in LC’s definition [2]. Additionally, different types of SARS-CoV-2 variants and existing immunity against COVID-19 can influence LC’s incidence or prevalence [13]. The definitions of LC used in studies, along with inclusion or exclusion criteria, study design and methods, and study setting (e.g., inpatient-based or outpatient-based), can significantly affect the effect size and outcomes of interventions as well [84]. Furthermore, LC symptoms appear to evolve over time; therefore, study outcomes may vary depending on the timing of the LC assessment after a COVID-19 diagnosis [84].

### 6.1. COVID-19 Vaccines Reduce the Risk of Long COVID

Except for two large staggered retrospective cohorts [49,61], most studies only included patients diagnosed with COVID-19 to assess the effect of vaccines—likely underestimating the true impact of vaccines as COVID-19 vaccines reduce the risk of infection and the severity of illness [85] and, consequently, LC. Nevertheless, most of these studies (ten out of fifteen) demonstrated that COVID-19 vaccines have a significant overall protective effect against developing LC.

There are several plausible reasons why some vaccine-related studies failed to show a protective effect. A booster compared to the primary series did not show a statistically significant protective effect in two included studies conducted during the Delta/Omicron and Omicron periods, respectively [57,66]. This finding is consistent with a recent meta-analysis [8]. In contrast, a cross-sectional study included in our review, conducted in Australia [63], found a dose–response benefit of vaccination against LC during the Omicron period. While further studies are needed to elucidate this, it is possible that booster vaccination may not provide substantial additional protection against LC beyond that provided by the primary vaccine series.

The unique study population and/or low vaccination rates due to vaccine unavailability during the early pandemic period may have reduced the statistical power of the following two included studies to detect a protective effect. Richard et al. [67] recruited young military personnel, only 22.9% of whom were fully vaccinated—partly due to the inclusion of individuals from the wild-type era and the younger age of the cohort. Of the participants in the study by Brunvoll et al. [48], which also included individuals from the wild type era, only 25% were vaccinated, likely because a significant portion of the participants were infected before vaccines became widely available.

Vaccination after COVID-19 infection is likely less protective against LC compared to vaccination before infection [8]. Most participants (73.6%) in the study by Babicki et al. [45] were vaccinated after COVID-19 infection, which may explain the lack of a protective effect.

Over time, LC rates have declined, associated with viral variants and vaccination, with the highest rate observed in the pre-Delta era, followed by the Delta era and the Omicron era [13]. Additionally, there are significant differences in LC rates and vaccine dose requirements for protection depending on the COVID-19 era [13]. Therefore, comparisons from different COVID-19 eras can introduce a selection bias and lead to erroneous results. For example, staggered cohorts [49,61] selected equivalent numbers of vaccinated and unvaccinated individuals from each era to avoid selection bias. However, studies [48,58,67] which included patients across different COVID-19 eras, including the wild-type era, may have disproportionately selected patients from the early COVID-19 eras, when vaccines were unavailable or vaccination rates were low, and LC rates were likely high. This selection method could have introduced a bias in the estimates.

Despite differences in patient populations, designs, methods, and the type of SARS-CoV-2 variant, COVID-19 vaccines are generally protective against LC. We found a stronger effect of vaccines when LC was defined as symptoms lasting 28 days or longer (Appendix A), which implies that COVID-19 vaccines reduce both acute COVID-19 illness and LC symptoms.

### 6.2. Equivocal Evidence for Protective Effect of Antivirals and Other Drugs

Studies have yielded mixed results, potentially due to differences in study design, patient population, study setting, the timing of antivirals, and possible measurement biases. Three studies (wild type and Alpha variant) on remdesivir demonstrated conflicting results; a randomized study demonstrated no effect [60], but a prospective study [78] and a retrospective case–control study demonstrated a protective effect on the development of LC [55]. The randomized study assessed LC symptoms 1 year after COVID-19, whereas the other two studies assessed them 12 weeks after COVID-19 or later. In addition to differences in the study design and patient population, the timing of symptom assessment may have contributed to different results.

Similarly, the results for nirmatrelvir/ritonavir also varied by study design, setting, and patient population. A prospective study of hospitalized patients in China [62] and a retrospective study of outpatients in Italy [46] demonstrated a protective effect of nirmatrelvir/ritonavir, whereas a prospective cohort study and a retrospective study in the US involving outpatients did not demonstrate a protective effect [51,53]. In the US, the Food and Drug Administration (FDA) approved the use of nirmatrelvir/ritonavir to treat mild to moderate COVID-19 to prevent severe infection [86], so nirmatrelvir/ritonavir has hardly been used in inpatient settings. While further studies on this topic are necessary to reach a more definite conclusion, inconsistent results indicate the uncertain effect of antivirals in preventing LC. It is possible that immune activation triggered by COVID-19, which contributes to the development of LC, may not be easily mitigated by antivirals.

A protective effect of antivirals against LC was more notable when LC was defined as symptoms lasting 28 days or longer, which indicates that antivirals may help reduce the symptoms of acute COVID-19 illness as symptoms of acute COVID-19 can last longer than 28 days.

A randomized study and a secondary analysis of a randomized study [56,65] did not demonstrate a protective effect of COVID-19 convalescent plasma transfusion on the development of LC in outpatient and hospitalized settings. However, a randomized study demonstrated a protective effect of metformin use in COVID-19 outpatients with overweight or obesity [47]. In addition, two retrospective studies demonstrated a protective effect of corticosteroid use in hospitalized COVID-19 patients on the development of LC [50,52]. Further studies would be valuable in understanding the underlying mechanisms and impact of these drugs on LC.

### 6.3. Effect of Nutrients and Lifestyle Factors on Long COVID

Besides vaccines, antivirals, and the above medications, other factors such as nutrients and lifestyle factors may play an important role in the development of LC. While a randomized study did not demonstrate a protective effect of multiple micronutrient supplements (MMSs) on LC incidence [83], it is possible that a longer duration of MMS use could yield different results. A prospective UK cohort study examining the association between modifiable lifestyle factors prior to COVID-19 and LC (with LC defined as symptoms lasting 30 days or longer) demonstrated that the risk of LC was reduced with a favorable lifestyle compared to an unfavorable lifestyle [87]. Additional studies on modifiable lifestyle factors to prevent LC are also warranted, especially since lifestyle modification can be potentially achievable at a low cost and with minimal adverse effects.

### 6.4. Limitations

Our study has several limitations. We might have missed some studies due to our inclusion criteria, although our review was comprehensive, extending to vaccine- and antiviral-related studies that defined LC as symptoms lasting 28 days or longer. There are no randomized trials to assess the impact of vaccines on the development of LC, although the majority of the studies demonstrate that vaccines have a robust protective effect. There is a limited number of studies for antivirals and other treatments; therefore, drawing a conclusion requires caution. The included studies assessed LC at different time points to determine the impact of vaccines, antivirals, or other treatments, which could have influenced the study outcomes. Due to heterogeneity in study designs and methods, statistical outcomes could not be uniformly reported.

## 7. Conclusions

Long COVID is a serious global burden that has impacted millions of individuals worldwide [2]. To the best of our knowledge, our study is the first systematic review addressing various preventive strategies for LC, particularly using the LC consensus definition.

Based on the data available at this time, COVID-19 vaccines likely reduce the risk of developing LC; however, evidence for antivirals and other drugs in preventing LC is still in its infancy. Future randomized, multicenter studies would offer valuable insights to (1) strengthen the evidence on the protection provided by COVID-19 vaccines against LC; (2) understand the impact of booster vaccines, antivirals, and other drugs on LC; and (3) understand the role of lifestyles in developing LC.

A lack of biomarkers that can assist in predicting or diagnosing LC hinders us from effectively preventing LC. Studies aiming to identify biomarkers that can predict the development of LC or diagnose LC are crucial to reduce the burden of LC. As our understanding of the pathophysiology of LC matures, our diagnostic, therapeutic, and preventive options will evolve, which will eventually reduce the morbidity associated with LC.

## Figures and Tables

**Figure 1 idr-17-00056-f001:**
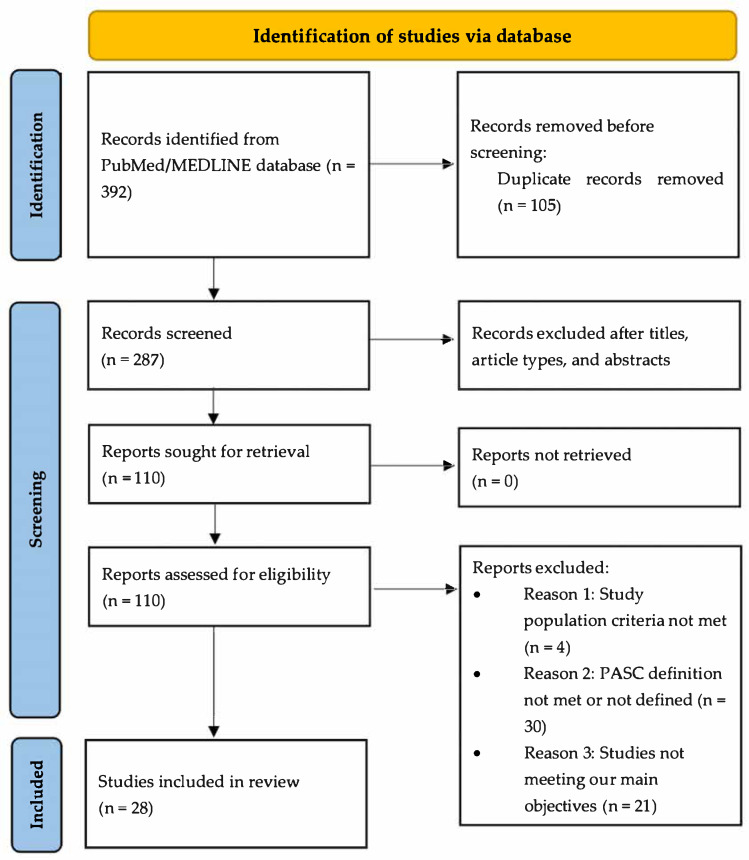
PRISMA diagram showing selection of studies for review.

**Table 1 idr-17-00056-t001:** Impact of COVID-19 vaccine on prevention of long COVID.

Study Author	Study Design/Country (Data)	Study Period/Participants (SARS-CoV-2 Variants)	LC Cases	Patient n (F %)	Age (Years), Mean ± SD or Median (IQR)	Vaccinated (%); Vaccine Type and Vaccination Time	Vaccine Impact on LC
Ayoubkhani et al. [44]	Prospective cohort/UK	Community based; visit during period of Feb–Sep 2021; COVID-19 at least 12 wks before final visit/(A, Δ)	LC 3–10 mo after COVID-19	COVID-19, n = 28,356 (F 55.6%)	45.9 ± 13.6	Vaccinated (100%): BNT162b2, mRNA-1273, or ChAdOx1 after COVID-19	Protective, 1st vaccine 12.8% reduction in odds (*p* < 0.001); 2nd vaccine provided additional 8.8% reduction (*p =* 0.003)
Antonelli et al. [66]	Prospective case–control study/UK	Community based; June–Nov 2021 (Δ) and Dec 2021–Apr 2022 (O)/cases (third dose recipient), controls (second dose recipient)	LC Sx ≥ 12 wks	All COVID-19, Delta: n = 1910 in each group (F 57%); Omicron: n = 7894 in each group (F 60.7%)	Delta: cases 64 ± 12.8, controls 63.7 ± 12.9; Omicron: cases 45.5 ± 16.3, controls 44.3 ± 17.7	Vaccinated (100%): 3 doses of monovalent in cases, 2 doses in controls) before COVID-19	No differences between cases and controls in both Δ and O eras, but trend towards protection in vaccinated individuals; when LC, Sx ≥ 4 wks, cases with aOR 0.56 (95% CI: 044–0.70) during Δ era
Richard et al. [67]	Prospective cohort/USA	Feb 2020-Dec 2021 (wild-type, A, Δ), data from MHS EPICC study	LC Sx ≥ 3 mo	COVID-19, n = 1832 (F 39%)	40.5 ± 13.7 (age range 18–44)	Fully vaccinated (22.9%) [2 doses of BNT162b2 or mRNA-1273 or 1 dose of Ad26.COV2.S] before COVID-19	Trend towards protective effect, RR 0.73 (95%CI: 0.47–1.14) ** (when LC Sx ≥ 28 d, RR 0.72 (0.54–0.96)
Nehme et al. [59]	Prospective longitudinal cohort/Switzerland	Outpatients, COVID-19 during period of Dec 2021–Feb 2022 (O)	LC 3 mo after COVID-19	COVID-19, n = 1807 (F 62.3%); COVID-19 negative, n = 882 (F 63.9%)	COVID-19 positive, 41.6 ± 13.5; COVID-19 negative, 43.7 ± 14.9	Fully vaccinated (75.5%) [at least 2 doses of mRNA-1273 (61.2%) or BNT162b2 (36.1%)] before COVID-19	Protective, adjusted prevalence 9.7% vs. 18.1% *(p* < 0.001)
Brunvoll et al. [48]	Prospective cohort (Norweigian COVID-19 cohort)/Norway	COVID-19 during period of Nov 2020–Oct 2021 (wild-type, A, Δ)	LC 3 mo-15 mo after COVID-19	COVID-19, n = 1420 (F 71%)	Vaccinated, 48.3 ± 11.4; unvaccinated, 45.7 ± 12.3	Fully vaccinated (25%) [at least 2 doses of mRNA vaccines at least 2 wks before COVID-19]	No differences in all components except for memory problem, fully vaccinated vs. unvaccinated, 11.9% vs. 17.3% (*p* = 0.02)
Abu Hamdh et al. [43]	Prospective cohort/Palestine	COVID-19 during period of Sep 2021– Jan 2022, with FU phone interviews on d 10, 30, 60, 90/(mainly Δ)	LC at 90 d	COVID-19, n = 669 (F 57%)	35.9 ± 11.5	Vaccinated (41%) [BNT162b2 (17.8%), Sputnik Light (12.7%), mRNA-1273 (3.7%), Sputnik V (2.4%), ChAdOx1 (2.4%)] before COVID-19	Protective, ≥1 dose vaccinated vs. unvaccinated, aOR 0.15 (95% CI: 0.09–0.24)
Fatima et al. [54]	Prospective cohort/Pakistan	Hospitalized patients with COVID-19 during period of Feb 2021–June 2021/(A)	LC at 12 mo	COVID-19 patients admitted to Aga Khan University hospital, n = 481 (F 38.3%)	56.9 ± 14.3	# Fully vaccinated (19%): 2 does of vaccines; partially vaccinated (19.2%): 1 dose before COVID-19	Protective, fully vaccinated aOR 0.38 (95% CI: 0.20–0.70), partially vaccinated aOR 0.44 (95% CI: 0.24–0.80)
Nascimento et al. [68]	Prospective cohort/Brazil	Hospitalized patients with COVID-19 during May 2021 and Feb 2022 (A, Δ, O)	LC at 90 d	COVID-19 patients, n = 412 (35.4%)	60 (IQR 48–72)	Fully vaccinated (44.9%) [1 dose of Janssen or 2 doses of # other vaccines before COVID-19]	Protective, aOR for fully vaccinated 0.55 (*p*= 0.007)
Català et al. [49]	Staggered retrospective cohort /UK (CPRD); Spain (SIDIAP); Estonia (CORIVA)	Primary care data; registered by Jan or Feb 2021, with FU until Jan 2022 (UK), June 2022 (Spain), Dec 2022 (Estonia)/(A, Δ, O)	LC between 90 d and 365 d after COVID-19	Over 10 million vaccinated vs. over 10 million unvaccinated (n/a)	n/a	Vaccinated with 1 dose (BNT162b2, ChAdOx1, mRNA-1273, or Ad26.COV2.S) +GOLD (49.7%); * AURUM (49.4); SIDIAP (51.5%); CORIVA (19.4%) before COVID-19	Protective: +GOLD, sHR 0.54 (95% CI: 0.44–0.67); * AURUM, 0.48 (0.34–0.68); SIDIAP, 0.71 (0.55–0.91); CORIVA, 0.59 (0.40–0.87)
Trinh et al. [61]	Staggered retrospective cohort/Norway (Norwegian linked health registries)	Primary care data; vaccination roll out from Jan to Aug 2021, FU up to 1 yr/(A, Δ, O)	LC between 90 d and 365 d after COVID-19	Over 2.3 million vaccinated vs. over 1.5 million unvaccinated (n/a)	n/a	Vaccinated with at least 1 dose (60.7%) [BNT162b2, mRNA-1273, or ChAdOx1] at least 14 d before COVID-19	Protective, sHR 0.64 (95% CI: 0.55–0.74)
Luo et al. [57]	Retrospective cohort/Hong Kong	Outpatient setting; COVID-19 during period of Dec 2021–May 2022/(mainly O)	LC at 6–12 mo after COVID-19	COVID-19, n = 6242 (F 66.9%)	47 (IQR 36–60)	Boosted (57.5%; 3 or more BNT162b2 or CoronaVac) vs. less than 3 doses before COVID-19	Not protective, aOR 1.11 (95% CI: 0.99–1.24)
MacCallum-Bridges et al. [58]	Population-based retrospective cohort/USA	Outpatient setting; COVID-19 during period of March 2020–May 2022/(wild-type, A, Δ, O)	LC at 90 d after COVID-19	COVID-19, n = 4695 (F 54.0%)	age 18–29 (25.7%); age 30–49 (37.6%); age 50–64 (24.2%); 65+ (13.5%)	Fully vaccinated (27.9%) [2 doses of BNT162b2, mRNA-1273, ChAdOx1, or Sinovac or 1 dose of Ad26.COV2.S] before COVID-19	Protective, aPR 0.42 (95% CI: 0.34–0.53)
Babicki et al. [45]	Retrospective cohort/Poland (STOP-COVID registry)	Unspecified study period, FU visits at 3 mo and 12 mo after COVID-19/(n/a)	LC at 1 yr after COVID-19	COVID-19, n = 801 (F 65.4%)	53.5 ± 12.8	Fully vaccinated (83%) [2 doses of BNT162b2, mRNA-1273, or ChAdOx1 or 1 dose of Ad26.COV2.S], 73.6% vaccinated after COVID-19, 9.4% before COVID-19	No differences except headache (17.4% vs. 29.4%, *p* = 0.001), arthralgia (5.4% vs. 10.3%, *p* = 0.032), and dysregulation of HTN (11.6% vs. 18.4%, *p* = 0.030) were more common in unvaccinated
Woldegiorgis et al. [63]	Cross-sectional survey/Australia	COVID-19 during period of July–Aug 2022, FU in 90 d/(O)	LC at 90 d after COVID-19	COVID-19, n = 11,697 (F 52.0%)	age 18–29 (20.9%); age 30–39 (21.0%); age 40–49 (18.7%); 50–59 (18.0%); 60–69 (11.6%); 70+ (9.7%)	#Vaccinated, 0–2 (6%); 3 doses (76.3%); 4 doses (17.7%) before COVID-19	Protective, 3 doses aRR 1.3 (95% CI: 1.1–1.5); 0–2 doses aRR 1.4 (1.2–1.8) compared to 4 or more vaccine doses
Xie et al. [64]	Cross-sectional survey/USA	Outpatient setting; 2022 National Health Interview Survey/(O)	LC 3 mo or longer post-infection	COVID-19, n = 8757 (weighted 87,509,670) (F 53.3%)	age 18–29 (23.8%); age 30–39 (21.3%); age 40–49 (18.2%); age 50–64 (23.2%); 65+ (13.5%)	# Vaccinated, 1 dose (17.3%); initial series (33.3%); booster (27.2%) before COVID-19	Protective, booster vs. unvaccinated, aOR 0.75 (95% CI: 0.61–0.93)

** Not statistically significant; # vaccine type not specified. Abbreviations; LC, long COVID; n, number; F, female; UK, United Kingdom; USA, United States of America; MHS EPICC, Military Health System Epidemiology, Immunology, and Clinical Characteristics of Emerging Infectious Diseases With Pandemic; CPRD, Clinical Practice Research Datalink; +GOLD, CPRD GOLD; * ARUM, CPRD ARUM; SIDIAP, Information System for Research in Primary care; FU, follow-up; d, days; wks, weeks; mo, months; yrs, years; n/a, not available; CI, confidence interval; *p*, *p*-value; aOR, adjusted odds ratio; sHR, sub-distribution hazard ratio; aPR, adjusted prevalence ratio; aRR, adjusted risk ratio; HTN, hypertension; Jan, January; Feb, February; Aug, August; Sep, September; Oct, October; Nov, November; Dec, December; A, Alpha; Δ, Delta; O, Omicron.

**Table 2 idr-17-00056-t002:** Impact of antivirals on the prevention of long COVID.

Study Author	Study Design/Country	Study Period/Participants/(SARS-CoV-2 Variants)	LC Cases	Patient n (F %)	Age, Mean ± SD or Median (IQR)	Antivirals (Treated %)	Antiviral Impact on LC
Nevalainen et al. [60]	Randomized trial/Finland	Hospitalized patients with COVID-19 during period of July 2020–Jan 2021/(wild type, A)	LC 1 yr after COVID-19	Treated, 114 (F 35.1%); untreated, 94 (F 36.2%).	Treated, 57.2 ± 13.5; untreated, 59.7 ± 13.2	Remdesivir (200 mg on 1st day, then 100 mg daily for maximum of 10 days)	No differences between remdesivir-treated and untreated with wide CI
Boglione et al. [78]	Prospective cohort/Italy	Hospitalized patients with COVID-19 during period of March 2020–Jan 2021(wild type, A)	LC symptoms ≥ 12 wks	Total of 449 (F 22%); Remdesvir-treated, 163; untreated, 165	65 (IQR 56–75.5)	Remdesivir	Protective, OR: 0.64 (95% CI: 0.41–0.78)
Fernández-de-las-Peñas et al. [55]	Retrospective, case–control study/Madrid, Spain	Hospitalized patients with COVID-19 during period of Sep 2020–March 2021/(wild type, A)	LC 3 mo or later following COVID-19	Treated, 216 (F 43.5%); untreated, 216 (F 43.5%)	Treated, 55.4 ± 12.6; untreated, 55.6 ± 12.7	Remdesivir (200 mg on 1st day, then 100 mg daily for maximum of 10 days)	Protective, OR: 0.401 (95 CI: 0.26–0.63)
Durstenfeld et al. [53]	Prospective cohort/USA	Vaccinated outpatients with their first SARS-CoV-2 positive result between March and Aug 2022/(O)	LC 90 d or later following COVID-19	Treated, 353 (F 53.3%); untreated, 1258 (F 64.9%)	Treated, 62.1 ± 12.7; untreated, 55.1 ± 13.6	Nirmatrelvir/ritonavir	No association, aOR: 1.15 (95% CI: 0.80–1.64)
Wang et al. [62]	Prospective cohort/Shanghai, China	Admitted with COVID-19 and then discharged between April and June 2022/(O)	LC 6 mo after being discharged	COVID-19, 634 (F 54.4%)	74.1 ± 11.4	Nirmatrelvir/ritonavir	Protective, OR: 0.35 (95 CI: 0.21–0.60)
Congdon et al. [51]	Retrospective cohort/New York, USA	Phone interviews between May and Nov 2022; COVID-19 four mo before the phone interview/(O)	LC 4 mo after COVID-19	Treated, 250 (F 66.4%); untreated, 250 (F 73.6%); hospitalized (1%)	50.6	Nirmatrelvir/ritonavir	No reduction in overall risk of LC (incidence of 44% vs. 50%, *p* = 0.21; aOR: 0.83, 95% CI: 0.57–1.2).
Bertuccio et al. [46]	Retrospective cohort/Italy	Outpatients with mild to moderate COVID-19 during period of April 2021–March 2022/(A, 3.5%; Δ, 2.2%; O 94.3%)	LC 3 mo after COVID-19	COVID-19, 649 (F 48.4%)	67 (IQR 54–76)	77 with antivirals (molnupiravir, 41.6%; nirmatrelvir/ritonavir, 13.0%; remdesivir, 45.5%); 141 with mAbs (bamlanivimab/ etesevimab, 44.7%; casirivimab, 16.3%; sotrovimab, 39.0%)	Antiviral protective, aOR of 0.43 (95% CI: 0.21–0.87) for any symptoms; mAbs protective, aOR of 0.48 (0.25–0.92) for neuro-behavioral symptoms

Abbreviations; LC, long COVID; n, number; F, female; USA, United States of America; FU, follow-up; d, days; wks, weeks; mo, months; yrs, years; mAbs, monoclonal antibodies; CI, confidence interval; *p*, *p*-value; aOR, adjusted odds ratio; Jan, January; Aug, August; Sep, September; Nov, November; A, Alpha; Δ, Delta; O, Omicron.

**Table 3 idr-17-00056-t003:** Impact of other treatments on prevention of long COVID.

Study Author	Study Design/Country (Data)	Study Period/Participants/ (SARS-CoV-2 Variants)	LC Cases	Patient n (F %)	Age (Years), Mean ± SD or Median (IQR)	Treatment (Treated %)	Treatment Impact on LC
Gebo et al. [56]	Randomized clinical trial/USA	Outpatients with COVID-19 between June 2020 and Oct 2021/(wild type, A, Δ)	LC 90 d after CCP	882 (F 57.4%)	43 ± n/a	CCP	No association, aOR: 0.75 (95% CI: 0.46–1.23)
Yoon et al. [65]	Secondary analysis of randomized clinical trial/USA (CONTAIN-RCT)	Hospitalized with COVID-19 between April 2020 and March 2021/(wild type, A)	LC 18 mo post-randomization	281 (F 44.5%)	59 (IQR 50–67)	CCP	No association, aOR: 0.95 (95% CI: 0.54–1.67)
Bramante et al. [47]	Randomized clinical trial/USA	Enrolled from Dec 2020 to Jan 2022/(wild type, A, Δ, O)	LC 10 mo after randomization	1125 (F 56%)	45 (IQR 37–54)	# Metformin, ivermectin, fluvoxamine	Only metformin was protective; HR: 0.58 (95% CI: 0.38–0.88); ivermectin, HR: 0.99 (0.59–1.64); fluvoxamine, HR: 1.36 (0.79–2.39)
Davelaar et al. [52]	Retrospective cohort/Netherlands	Hospitalized with COVID-19 between March 2020 and Sep 2021/(wild type, A, Δ)	LC 6 mo after being discharged	123 (F 38.2%)	62.1 ± 9.5	Corticosteroids	Protective, aOR: 0.32 (95% CI: 0.11–0.90)
Catalán et al. [50]	Retrospective cohort/Spain	Telephone survey between March 2021 and April 2021 for patients hospitalized with COVID-19 one yr earlier/(wild type)	LC 1 yr after being discharged	76 (F 38%)	Treated, 68.5 (IQR 60.2–75.5); untreated, 61.5 (IQR 52.7–72.5)	Corticosteroids	Protective: headache (6.3% vs. 25%, *p =* 0.032); dysphagia (11.4% vs. 0%, *p* = 0.049); depression (22.7% vs. 3.1%, *p* = 0.016), chest pain (11.4% vs. 0%, *p*= 0.049); bodily pain (* SF-36: 100 vs. 75, *p* = 0.017), mental health (* SF-36: 86 vs. 76, *p* = 0.027)
Tomasa-Irriguible et al. [83]	Randomized clinical trial/Catalonia, Spain	Outpatients with COVID-19 between Sep 2021 and Feb 2023/(Δ, O)	LC after 6 mo	246 (F 68.3%)	46.8 ± 16.3	Multiple micronutrient supplement	No reduction in incidence of LC (intervention, 27.7% vs. placebo, 25%; *p* = 0.785)

# Metformin: a dose of 500 mg on day 1, and then 500 mg twice daily for 4 days, and then 500 mg in the mornings and 1000 mg in the evenings for 14 days; ivermectin: a dose of 390–470 mcg/kg/day for 3 days; fluvoxamine: a dose of 50 mg on day 1, and then 50 mg twice daily for 14 days. * SF-36: Short Form-36 quality of life questionnaire. Abbreviations; LC, long COVID; n, number; F, female; USA, United States of America; FU, follow-up; d, days; wks, weeks; mo, months; yr(s), year(s); n/a, not available; CCP, COVID-19 convalescent plasma; CI, confidence interval; *p*, *p*-value; aOR, adjusted odds ratio; HR, hazard ratio; Jan, January; Feb, February; Sep, September; Oct, October; Dec, December; A, Alpha; Δ, Delta; O, Omicron.

## Data Availability

No new data were created or analyzed in this study. Data sharing is not applicable to this article.

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
