# Peer review of "Long COVID: A Systematic Review of Preventive Strategies"

_2036-7449, 2025, doi:10.3390/idr17030056_

Round 1

Reviewer 1 Report

Comments and Suggestions for Authors

IDR-3575952

  1. Introduction

  • The introduction provides a thorough analysis of the characteristics and mechanisms associated with the onset of Long COVID. However, given the explosion of scientific literature on COVID-19—often characterized by redundant publications—it would be useful to include a more detailed comparison with findings from other systematic reviews (e.g., doi:10.1016/j.cmi.2024.07.006), in order to contextualize the current work within the broader research landscape
  • It would beneficial if you condense the introduction, avoiding repetitions

  1. Methods

  • Are the authors sure that they included all the studies in literature? According to some quick research, some studies are missing (e.g. https://doi.org/10.1186/s12879-024-10226-1). In case inclusion criteria are not met in your opinion, it would be beneficial to mention the reasons in the discussions.
  • In contrast to the introduction, the Methods section appears rather underdeveloped, suggesting limited effort in its preparation. The authors adopted a descriptive approach, which is acceptable; however, they did not justify this choice. Even if a meta-analysis was not feasible, a narrative synthesis stratified by subgroups would have enhanced both clarity and interpretability of the findings
  • I cannot find the risk of bias assessment
  • As I understand, the absence of a meta-analysis may be justified by the heterogeneity of the available data. However, it remains unclear why the review was not registered on PROSPERO or any other platform (OSF, research registry). Given that registration enhances transparency, avoids duplication, and strengthens the methodological rigor of systematic reviews—even in the absence of a meta-analysis—an explanation for this omission would be advisable

  1. Discussion

  • Synthesize findings more than narrating; reduce redundancy. While I could provide specific examples, it appears that this descriptive approach has been applied consistently throughout the Discussion section. Given that the results are largely self-explanatory, the authors should focus more on comparing their findings with existing literature and similar experiences. As it stands, the mere description of results fails to capture the reader’s attention or add interpretative value.
  • The section '6.3. Nutrients and Lifestyle Factors on Long COVID' is certainly interesting; however, it may not warrant such extensive discussion, as it does not represent the primary focus of this review. A more concise treatment would help maintain alignment with the main objectives of the study.

Reviewer 2 Report

Comments and Suggestions for Authors

This is a well-written and important review about long-covid (LC) and how vaccines, antivirals and corticoids can prevent it. I only have a few comments to improve it:

The authors cite several “gastrointestinal, immune dysregulation, cardiovascular, endocrine, etc” symptoms associated with LC. Please exemplify what these symptoms are for each.

Since immune dysregulation is an important part of long covid, the review would benefit from a more in depth discussion on the immune responses against covid. For example, discussing also how exhausted T cells might promote LC. Are the types of HLAs associated with LC?

In lines 120-127, the authors state that LC may be caused by systemic inflammation. How? Bystander activation of immune cells? Epitope spreading? Please discuss.

The number in the tables after “n=” are not standardized, sometimes they are “1,740”, sometimes “1740”. Please correct.

In the study of Antonelli [64], the vaccines used offered protection against delta and omicron? If not, please discuss the results in light of this fact.

Reviewer 3 Report

Comments and Suggestions for Authors

The manuscript is well-written and addresses a highly relevant and timely topic. However, several important issues should be addressed to enhance the clarity, consistency, and scientific rigor of the work prior to publication:

  1. The manuscript references multiple definitions of Long COVID as outlined by the WHO, NASEM, and CDC. However, it is unclear which definition was consistently applied during the study selection process. Clearly specifying and adhering to a single operational definition throughout the review would improve the study's methodological robustness.

  2. Although the authors have presented a PRISMA flow diagram, the manuscript lacks a formal risk of bias assessment or quality evaluation of the included studies — a critical component of PRISMA guidelines. Including this assessment would substantially improve the transparency and reliability of the review’s findings.

  3. The data presentation throughout the results section lacks uniformity. Vaccine, antiviral, and treatment outcomes are reported using a mixture of percentages, odds ratios, and adjusted ratios without a standardized format. Harmonizing the reporting style would greatly improve comparability and reader comprehension.

  4. The discussion briefly acknowledges demographic and variant-related diversity (such as military, hospitalized, and outpatient populations), but does not explore how these differences may have influenced the observed outcomes. A deeper analysis of this heterogeneity would enhance the interpretative strength of the review.

  5. Despite the availability of considerable quantitative data, the authors did not perform a meta-analysis. Including even a basic meta-analysis — for example, pooled odds ratios or a forest plot — would provide a clearer and more statistically supported synthesis of the available evidence.

  6. While the manuscript notes that some studies did not demonstrate protective effects of vaccines or antivirals against Long COVID, the discussion offers limited exploration of potential reasons for these discrepancies. A more thorough critical appraisal — including considerations of study design, variant-specific dynamics, and possible measurement biases — would strengthen the review’s conclusions.

Round 2

Reviewer 1 Report

Comments and Suggestions for Authors

No further comments.